# Wearable Noninvasive Glucose Sensor Based on Cu_x_O NFs/Cu NPs Nanocomposites

**DOI:** 10.3390/s23020695

**Published:** 2023-01-07

**Authors:** Zhipeng Yu, Huan Wu, Zhongshuang Xu, Zhimao Yang, Jian Lv, Chuncai Kong

**Affiliations:** Ministry of Education Key Laboratory for Non-Equilibrium Synthesis and Modulation of Condensed Matter, Shaanxi Province Key Laboratory of Advanced Functional Materials and Mesoscopic Physics, School of Physics, Xi’an Jiaotong University, Xi’an 710049, China

**Keywords:** noninvasive wearable sensing, metal oxide, sweat glucose detection

## Abstract

Designing highly active material to fabricate a high-performance noninvasive wearable glucose sensor was of great importance for diabetes monitoring. In this work, we developed Cu_x_O nanoflakes (NFs)/Cu nanoparticles (NPs) nanocomposites to serve as the sensing materials for noninvasive sweat-based wearable glucose sensors. We involve CuCl_2_ to enhance the oxidation of Cu NPs to generate Cu_2_O/CuO NFs on the surface. Due to more active sites endowed by the Cu_x_O NFs, the as-prepared sample exhibited high sensitivity (779 μA mM^−1^ cm^−2^) for noninvasive wearable sweat sensing. Combined with a low detection limit (79.1 nM), high selectivity and the durability of bending and twisting, the Cu_x_O NFs/Cu NPs-based sensor can detect the glucose level change of sweat in daily life. Such a high-performance wearable sensor fabricated by a convenient method provides a facile way to design copper oxide nanomaterials for noninvasive wearable glucose sensors.

## 1. Introduction

To control the evolution of diabetes mellitus and its dangerous consequences, it is crucial to constantly monitor the in-vivo and in-vitro glucose concentration in the blood or other bodily fluids [1,2]. Beyond using human blood as the analyte for invasive glucose sensing, sweat is also considered a promising target to indicate the blood glucose level owing to the non-invasive epidermal sensing [3,4]. Due to their excellent selectivity, strong anti-interference performance and high compatibility, glucose oxidase and glucose dehydrogenase electrochemical sensing technologies have been the primary foundation for human glucose sensors up to this point [5]. But pH, temperature, and dissolved oxygen all played a significant role in how active enzymes were.

Due to their stability and superior detection capability, several metal materials, especially gold [1,6,7,8], platinum [9], copper [10,11] and nickel [12,13,14], have gained a lot of emphasis in non-enzymatic glucose sensors [15]. Noble metal materials are pricey, which has hindered their utilization in non-enzymatic glucose sensors [16]. Despite the fact that inexpensive transition metals and their alloys had glucose-sensing properties, surface poisoning proved a significant issue [17]. Copper oxides deliver excellent electrochemical activity [18] and own a sufficient potential for non-enzymatic glucose oxidation [19,20,21,22,23]. Cu(II) oxide and Cu(I) oxide are widely known copper oxides that have outstanding sensing capacities. An electron pair reaction involving the redox couple ions Cu(III) and Cu(II) formed the foundation for the sensing mechanism [24,25]. The continuous transformation of electrons was available by the redox pair ions, which serve as an intermediary mediator [26,27,28,29]. The copper oxide function layer allows the glucose to transfer its released electrons to the current collector [30,31,32].

An abundance of active sites for the electrocatalytic reaction can be provided by a large surface area without the cover of surfactants, thus improving the sensor’s sensitivity. This is even more valuable when considering the low glucose concentration in human sweat [33]. Our previous work used H_2_O_2_ as the oxidant to directly grow the high surface area Cu_2_O/CuO nanosheets on the surface of a copper foil [34,35]. The surface of these nanosheets is active for glucose sensing, as no organic surfactant was involved in the reaction. However, the capability for H_2_O_2_ to oxide the Cu foil is limited and a high temperature (180 °C) is required to enhance the oxidation reaction [34]. When the Cu foil was replaced by the Cu nanowires, the generation of nanosheets can be performed at room temperature, but the reaction requires an extended duration [35]. The novel strategy was required to improve the oxidation capability of H_2_O_2_ to produce highly glucose-sensitive copper oxide materials for human sweat detection.

In this work, we synthesized Cu_x_O NFs/Cu NPs nanocomposites with abundant active sites with the help of CuCl_2_ and then evaluated their ability for noninvasive wearable glucose sensing. The schematic representation of the synthesis process and its application mechanism in noninvasive glucose detection was demonstrated in Figure 1. The involvement of CuCl_2_ can strengthen the oxidation of Cu nanoparticles to Cu_x_O nanoflakes at room temperature. The Cu_x_O NFs/Cu NPs nanocomposites based noninvasive wearable glucose sensor delivers high sensitivity, good stability, anti-inferring capability, and durability to mechanical deformation, enabling its application in monitoring glucose level change in human sweat. Our work demonstrates that the as-prepared Cu_x_O NFs/Cu NPs nanocomposites deliver high performance in sweat glucose sensing, enriching the application of metal oxide in wearable electrochemical sensors.

## 2. Materials and Methods

### 2.1. Chemicals and Reagents

Commercial Cu nanoparticles (NPs), CuCl_2_·2H_2_O, NaOH, NaCl, glucose, ascorbic acid (AA), Uric acid (UA), Urea, Nafion, KCl, K₃[Fe(CN)₆], K₄Fe(CN)₆, isopropanol and ethanol were purchased from Aladdin Co. and were used as-received without further purification (Analytically Reagent, AR, USA). BOPP waterproof tape (Box Sealing Tape-375) were purchased from 3M China Co., Ltd.. PU film (Medical no sensitive breathable tape, PU film-type C) were purchased from Shanghai Hons Medical Technology Co., Ltd. (Shanghai, China). Deionized water (18.4 MΩ cm^−1^, 23 °C) was used for all solutions preparation.

### 2.2. Instruments

A Bruker-AXS D8 ADVANCE diffractometer (Cu, kα = 0.1506 nm) was used to perform the X-ray diffraction (XRD) measurement. Thermo Fisher ESCALAB Xi+ (Al, Kα) was used for X-ray photoelectron spectroscopy (XPS). JSM-7000F field-emission scanning electron microscope (FE-SEM, JEOL, 15 kV) and JEM-2100 transmission electron microscope (TEM, JEOL, AC 120 kV) were used to assess morphology and HRTEM images, respectively. For electrochemical measurements, a CHI-660e electrochemical workstation was used.

### 2.3. Preparation of the Cu_x_O NFs/Cu NPs Nanocomposites

In a typical procedure, 0.1 g of Cu NPs was dispersed in DI water (10 mL) by 30 min ultrasonication. 1 mL of CuCl_2_·2H_2_O aqueous solution (0.1 M) was added to the suspension and stirred for 15 min. The obtained products were then centrifuged, washed with absolute alcohol and deionized water three times and dried at a 60 °C vacuum atmosphere for 2 h. Then 0.1 g of dried product was added in DI water (5 mL) and stirred for 30 min. 0.1 M H_2_O_2_ aqueous solution (10 mL) was added to the suspension and with further string for 2 h. The final products were then centrifuged, washed with absolute alcohol and deionized water three times, then dried at a 60 °C vacuum atmosphere for 10 h.

### 2.4. Preparation of the CuO NFs/Cu NPs Nanocomposites

0.1 g of Cu NPs was added into DI water (5 mL) and stirred for 30 min. 0.1 M H_2_O_2_ aqueous solution (10 mL) was added to the suspension and follower by 2 h stirring. The obtained products were then centrifuged, washed with absolute alcohol and deionized water three times, then dried at a 60 °C vacuum atmosphere for 10 h.

### 2.5. Preparation of the Cu_x_O NFs/Cu NPs Nanocomposites-Based Wearable Sensor

The Cu_x_O NFs/Cu NPs nanocomposites-based noninvasive wearable sensor was fabricated on the screen-printed electrodes with Ag/AgCl as the reference electrode, and carbon as both the counter electrode and the working electrode. The substrate is a polyethylene terephthalate film. Typically, 1 mg samples were dispersed in a solution composed of 250 μL isopropanol, 250 μL DI water mixed and 15 μL Nafion solution (isopropanol solution, 5%). The suspension was further mixed under ultrasonic agitation for 2 h and then 2 μL of the mixture was dropped onto the working electrode (diameter is 3 mm), followed by drying at room temperature for 6 h. For wearable application, the intergrade electrode was attached to the BOPP waterproof tape and following covered by PU film.

### 2.6. Electrochemical Measurements

All the electrochemical experiments were conducted using a three-electrode electrochemical system. The electrolyte is a 100 μL aqueous solution of NaOH (0.1 M, pH~13) and NaCl (0.1 M). Cyclic Voltammograms (CVs) were recorded between 0 and 0.7 V. CVs were collected between 0 and 0.1 V for ECSA measurement. The electrolyte used in Electrochemical Impedance Spectroscopy (EIS) measurements is a combination of 0.1 KCl, 0.1 M K_3_[Fe(CN)_6_] and 0.1 M K_4_Fe(CN)_6_.

## 3. Results and Discussion

### 3.1. Characterization of Materials

After the successive treatment of CuCl_2_ aqueous solution and H_2_O_2_ aqueous solution, Cu NPs were transferred to the intermediate product of Cu_2_(OH)_3_Cl/CuCl/Cu NPs nanocomposites and the final products are Cu_x_O NFs/Cu NPs nanocomposites. The morphology of Cu_x_O NFs/Cu NPs nanocomposites was characterized by FESEM (Figure 2a), TEM (Figure 2b), HRTEM (Figure 2c,d), line profile (Figure 2 insert c1,d1) and FFT (Figure 2 insert c2,d2). The FESEM and TEM show the as fabricated products mainly consist of nanoflakes, which deliver a high surface area for lateral glucose sensing. The interplanar distances of 0.2465 nm and 0.2381 nm measured from HRTEM images (Figure 2c,d) belong to the (111) and (111) planes of Cu_2_O and CuO, respectively. As a comparison, the CuO NFs/Cu NPs nanocomposites without the involvement of CuCl_2_ treatment show a similar morphology with that of Cu_x_O NFs/Cu NPs nanocomposites (Appendix A), suggesting that the H_2_O_2_ is the key step to generate the nanoflakes structure. In addition, as shown in Appendix A, the intermediate product of Cu_2_(OH)_3_Cl/CuCl/Cu NPs nanocomposites still show the solid particle shape.

The crystal structure of Cu_x_O NFs/Cu NPs nanocomposites was measured by XRD. The pattern of Cu_x_O NFs/Cu NPs nanocomposites (Figure 2e) have the peaks of CuO (marked with plus sign, JCPDS No. 44-0706), Cu_2_O (marked with minus sign, JCPDS No. 05-0667) and Cu (marked with asterisk sign, JCPDS No. 04-0836). The peaks of Cu_x_O NFs/Cu NPs nanocomposites at 43.4°, 50.5° and 74.2° corresponding to lattice plane (111), (200) and (220) of Cu, suggesting that Cu NPs are still not fully transformed into the oxides during the treatment of CuCl_2_ and the subsequent oxidation of H_2_O_2_. However, compared to the CuO NFs/Cu NPs nanocomposites, the peaks of Cu are much weak, indicating the introduction of CuCl_2_ can enhance the following oxidation reaction to generate more copper oxide nanoflakes as shown in the XRD pattern of the intermediate product of Cu_2_(OH)_3_Cl/CuCl in Appendix A, the pattern can be attributed to Cu_2_(OH)_3_Cl (marked with equal sign, JCPDS No. 86-1391), CuCl (marked with pound sign, JCPDS No. 06-0344) and Cu (marked with asterisk sign). The Cu_2_(OH)_3_Cl/CuCl/Cu NPs nanocomposites were synthesized by Cu NPs following treatment with CuCl_2_ aqueous solution, as shown in Appendix A, Figure 2e and Appendix A. In the subsequent oxidation by H_2_O_2_, these intermediate products were transferred to Cu_2_O and CuO nanoflakes.

To further investigate the valence state of the synthesized materials, XPS spectra (Figure 2f) of Cu_x_O NFs/Cu NPs nanocomposites were carried out. In Cu_x_O NFs/Cu NPs nanocomposites, the peaks located at 934.6 eV and 953.7 eV were assigned to the Cu 2p_3/2_ and Cu 2p_1/2_ of CuO, suggesting the presence of CuO [36]. The existence of shake-up satellite peaks at higher binding energy around 941.2 eV also confirmed the CuO state. Meanwhile, there were doublet peaks at 932.84 eV and 955.2 eV which were assigned to the Cu 2p_3/2_ and Cu 2p_1/2_ of Cu_2_O [34]. Only peaks belonging to Cu (932.3 eV and 952.1 eV) and CuO (933.1 eV and 953.0 eV) were found in the XPS spectra of commercial Cu NPs and the sample without the involvement of CuCl_2_ [23]. The XPS results further suggest that the CuO/Cu_2_O layer was formed on the surface of Cu NPs.

### 3.2. Electrochemical Measurements

The electrochemical measurement was performed on flexible sensor chips, as shown in Figure 3a. The CV characteristics of the bare Nafion, Cu NPs, Cu_2_(OH)_3_Cl/CuCl/Cu NPs nanocomposites, Cu_x_O NFs/Cu NPs nanocomposites and CuO NFs/Cu NPs nanocomposites were measured to compare their electrocatalytic performance (Appendix A). As shown in Figure 3b, the current response of curves of Cu_x_O NFs/Cu NPs nanocomposites is much higher than that of bare Nafion, Cu NPs, Cu_2_(OH)_3_Cl/CuCl/Cu NPs nanocomposites and CuO NFs/Cu NPs nanocomposites with the presence of 1.0 mM glucose. The CuCl_2_ transfer more Cu to generate Cu_2_O/CuO NFs and thus the response was much higher than that of the CuO NFs/Cu NPs nanocomposites. The response of the Cu_x_O NFs/Cu NPs nanocomposites-based sensor gradually raised towards the oxidation of glucose when the concentration was increased to 2.5 mM and the catalytic current peak ranges from 0.40 V to 0.60 V (Figure 3c). The CV property of Cu_x_O NFs/Cu NPs nanocomposites with different scan rates (v) was tested from 5 mV s^−1^ to 50 mV s^−1^ with the presence of 1.0 mM glucose concentration. The CVs of Cu NPs, CuO NFs/Cu NPs nanocomposites, Cu_2_(OH)_3_Cl/CuCl/Cu NPs nanocomposites, Cu_x_O NFs/Cu NPs nanocomposites with varied scan rates were shown in Appendix A for studying the electrochemical active surface area (ECSA). The extracted calibration curves of current response versus the scan rates with different samples for ECSA was shown in Appendix A. The ECSA of Cu NPs, Cu_2_(OH)_3_Cl/CuCl/Cu NPs nanocomposites, CuO NFs/Cu NPs nanocomposites and Cu_x_O NFs/Cu NPs nanocomposites were 118.51 μF·cm^−2^, 167.07 μF·cm^−2^, 50.8 μF·cm^−2^ and 140.05 μF·cm^−2^, respectively. In addition, the Electrochemical Impedance Spectroscopy (EIS) of Cu NPs, Cu_2_(OH)_3_Cl/CuCl/Cu NPs nanocomposites, CuO NFs/Cu NPs nanocomposites and Cu_x_O NFs/Cu NPs nanocomposites was measured and fitted in Appendix A. As shown in Appendix A, the transfer resistance value of Cu NPs, Cu_2_(OH)_3_Cl/CuCl/Cu NPs nanocomposites, CuO NFs/Cu NPs nanocomposites and Cu_x_O NFs/Cu NPs nanocomposites were 506.2 ohm, 339.6 ohm, 787.3 ohm and 198.7 ohm, respectively. This result demonstrated that, after the CuCl_2_ aqueous solution treatment and H_2_O_2_ aqueous solution, the ECSA was notably increased and transfer resistance value was highly reduced. As shown in Appendix A, the oxidation peak current of the as-prepared sensor rises with the scan rate and the corresponding fitting equation is I = 11.34 × v^0.5^ − 35.49. The linear relationship between the response current intensity and the square root of the scan rate suggests that the adsorption of glucose determined the redox reaction [34].

The amperometric curves of the Cu_x_O NFs/Cu NPs nanocomposites-based sensor at different potentials upon increased glucose concentration are shown in Appendix A. Figure 3d shows the extracted curves of a current response versus the glucose concentration at different potentials. It is found that the current responses at +0.40 V, +0.45 V, +0.5 V, +0.55 V and +0.6 V are lower than that at +0.55 V. So, the following amperometric curves were tested at +0.55 V. To calculate the sensitivity of the sensor, the amperometric responses of Cu_x_O NFs/Cu NPs nanocomposites-based sensor at +0.55 V with the increased glucose concentration were measured and the calibration curve between glucose concentration and the current was shown in Figure 3e,f, respectively. The sensitivity of the as-prepared sensor was 779 μA mM^−1^ cm^−2^ based on the linear fitting equation (I(μA) =9.93 + 0.055 × C (μM)), which was 4.0-fold that of CuO NFs/Cu NPs nanocomposites (196 μA mM^−1^ cm^−2^) (Appendix A). The relationship between current and glucose concentration follows the Michaelis-Menten equation. The response current intensity j could be expressed by j = K_cat_ × [S]/(Km + [S]) [34], in which the K_cat_, [S], K_m_ are substrate concentration, Michaelis constant, apparent catalytic turnover rate, respectively. In an extreme case, if the glucose concentration was at a low range and the K_m_ was larger than [S], the response current intensity j exhibits linear relation with the concentration of glucose. The detection limit was calculated by 3σ/s and the σ and s are defined as the background current’s standard deviation and the slope of the calibration curve, respectively. The detection limit of the Cu_x_O NFs/Cu NPs nanocomposites-based sensor was 79.1 nM. Table 1 shows the performance comparison among the Cu_x_O NFs/Cu NPs nanocomposites-based sensor and several reported non-enzyme glucose sensors. The as-prepared Cu_x_O NFs/Cu NPs nanocomposites-based sensor exhibits better overall performances than other sensors for more abundant active sites from nanoflake structure.

Figure 4a shows the amperometric response stability of Cu_x_O NFs/Cu NPs nanocomposites sensor at 1.0 mM glucose during 1000 s testing and no significant attenuation was detected. The amperometric responses of the sensor were measured 6 times and no remarkable deviation among each measurement, suggesting the great repeatability of our Cu_x_O NFs/Cu NPs nanocomposites-based sensor (Figure 4b). The anti-interference performance was measured by injecting 1.0 mM glucose first and then subsequently adding 10 mM urea, ascorbic acid (AA) and uric acid (UA). As shown in Figure 4c, signal responses of the sensor toward interfering chemicals were quite weak. In comparison, the sensor generated a strong current response when glucose was reinjected. The intensity of the second responding current was close to that of the first one. Figure 4d shows the normalized current response of different interfering species and the current response to 1.0 mM glucose was set as 100%. Cu_x_O NFs/Cu NPs nanocomposites-based sensors exhibited excellent anti-interference performances. These results suggest that the Cu_x_O NFs/Cu NPs nanocomposites-based sensor was reliable and suitable for the following noninvasive wearable glucose sensing.

### 3.3. Real Sample Analysis

The Cu_x_O NFs/Cu NPs nanocomposites-based sensor not only shows superior electrochemical performance but also delivers good flexibility, making it suitable for wearable applications. To evaluate the flexibility of our sensor, we test the sensing performance after repeated bending and twisting conditions. In our experiment, the sensor was twisted and bent to 90°, and then unfolded to restore to the original geometry, as shown in Figure 5a,b. After the deformation was completed, the sensor returned to its original state. The durability of mechanical deformation was assessed by checking the response to a glucose solution. After 100, and 500 cycles, respective amperometric tests were performed. Figure 5c,d show the amperometric responses of the sensor toward 0.5 mM glucose before and after cycling twisting and bending, respectively. The amperometric responses demonstrate that the sensor exhibits robust performance against repeated mechanical deformation, with insignificant departure from the original condition. The flexibility ensures good contact between the sensor and human skin and durability to mechanical deformation guarantees reliable sensing.

To further demonstrate the effectiveness of the Cu_x_O NFs/Cu NPs nanocomposites-based glucose sensor for wearable monitoring, the sensor was mounted on the arm of one volunteer to detect glucose concentrations in human sweat during the exercise (Figure 5e). In this study, a healthy subject (32 years old) who has no medical history of diabetes participated in the experiment. Before the mounting of the sensor, disinfection and cleansing were performed on the arm skin. The PU film was used to protect the sensor and keep the measurement condition with a steady wire connection. Then the volunteer was asked to do 30 min rope skipping to secrete sweat. The sweat sample was measured in situ without any treatment. As shown in Figure 5f, the sweat glucose concentrations of the volunteer changed before (−30 min) and after (+30 min, +3 h and +5 h) remarkably. The measured current response before a meal was lower than after a meal, suggesting that the glucose level in the subject’s sweat increased. Although the detection current has slight fluctuation, the obtained results show that the sensor signal increased when the glucose concentration of sweat was changed, promising the application of wearable Cu_x_O NFs/Cu NPs nanocomposites-based sensors in detecting the glucose concentration in human sweat sensors.

## 4. Conclusions

In conclusion, the Cu_x_O NFs/Cu NPs nanocomposites with a high surface area were successfully fabricated through the involvement of CuCl_2_. Its application as non-enzyme noninvasive wearable glucose sensor was evaluated. The as-prepared sensor electrode exhibits high sensitivity (779 μA mM^−1^ cm^−2^), wide linear range (higher than 2.5 mM), low detection limit (79.1 nM), durability to twisting and bending deformation and good selectivity. The sensitivity of Cu_x_O NFs/Cu NPs-based sensor was 3 times higher than that of the sensor based on CuO NFs/Cu NPs nanocomposites synthesized without CuCl_2_. The detection of glucose concentration in human sweat samples was successfully measured, indicating its potential application as a noninvasive wearable glucose sensor.

## Figures and Tables

**Figure 1 sensors-23-00695-f001:**
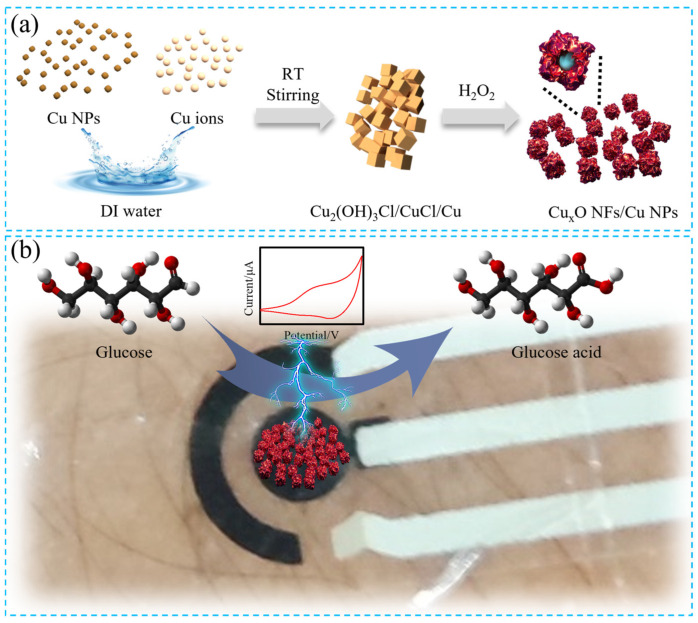
(**a**) Schematic illustration of the synthesis process of Cu_x_O NFs/Cu NPs nanocomposites and (**b**) their application mechanism in noninvasive glucose detection.

**Figure 2 sensors-23-00695-f002:**
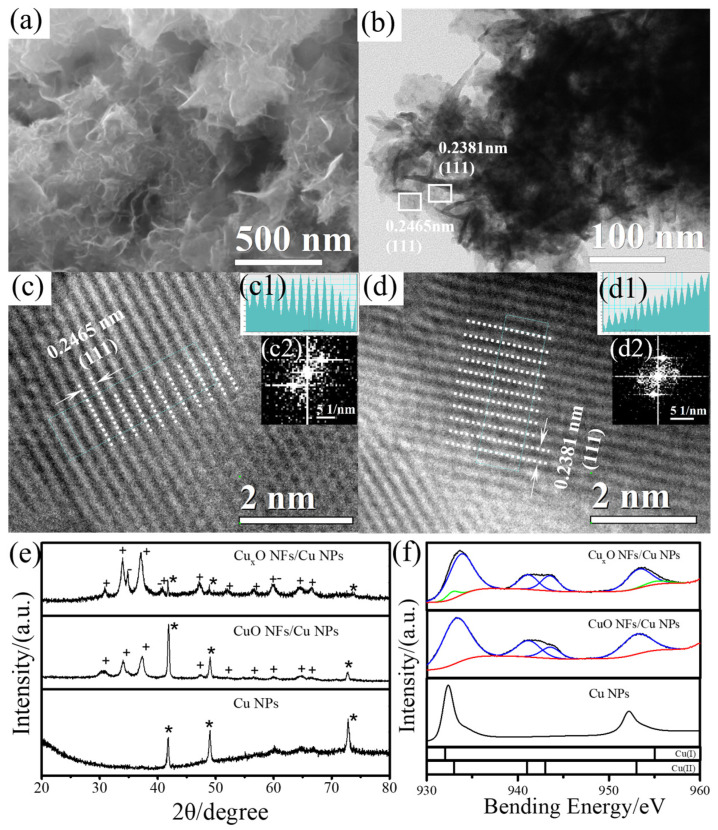
Images of (**a**) FESEM, (**b**) TEM, (**c**,**d**) HRTEM, (insert **c1**,**d1**) line profile and (insert **c2**,**d2**) FFT of Cu_x_O NFs/Cu NPs nanocomposites. (**e**) XRD patterns (the signs of “*”, “+”,”−” represent the phase of “Cu”, “CuO”, “Cu_2_O”, respectively) and (**f**) XPS Cu 2p spectra of Cu NPs, CuO NFs/Cu NPs nanocomposites and Cu_x_O NFs/Cu NPs nanocomposites.

**Figure 3 sensors-23-00695-f003:**
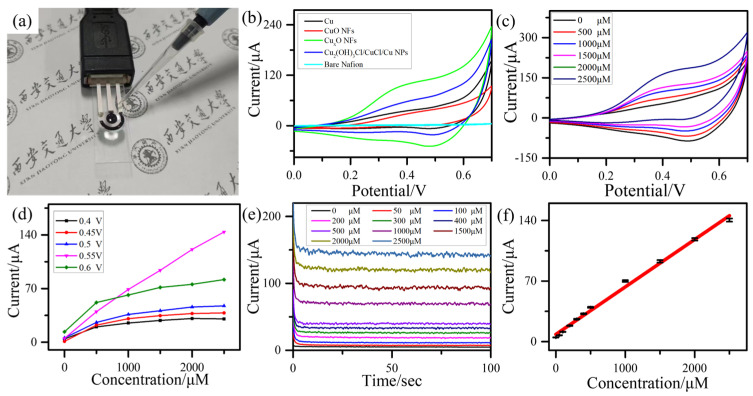
(**a**) Optical images of electrochemical measurement. (**b**) CV curves of different samples with 1.0 mM glucose concentration. (**c**) CV curves of Cu_x_O NFs/Cu NPs nanocomposites-based sensor with increased glucose concentration. (**d**) the extracted calibration curves of a current response versus the glucose concentration at different potentials from +0.4 V to +0.6 V. (**e**) Amperometric responses with increased glucose concentration on +0.55 V. (**f**) The extracted currents upon different concentrations and corresponding calibration curves.

**Figure 4 sensors-23-00695-f004:**
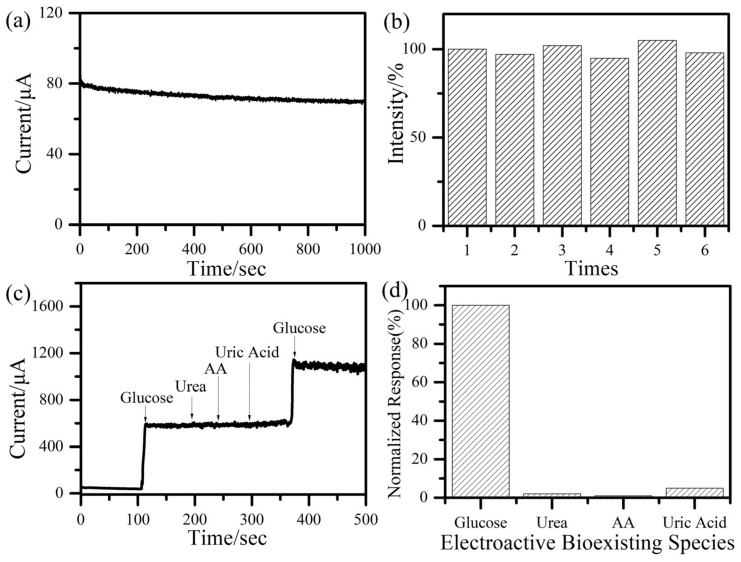
(**a**) The amperometric response with 1.0 mM glucose for 1000 s. (**b**) normalized amperometric response during the repeating test. (**c**) amperometric response of the sensor during consecutive addition of interference chemicals at +0.55 V. (**d**) normalized response of interferences.

**Figure 5 sensors-23-00695-f005:**
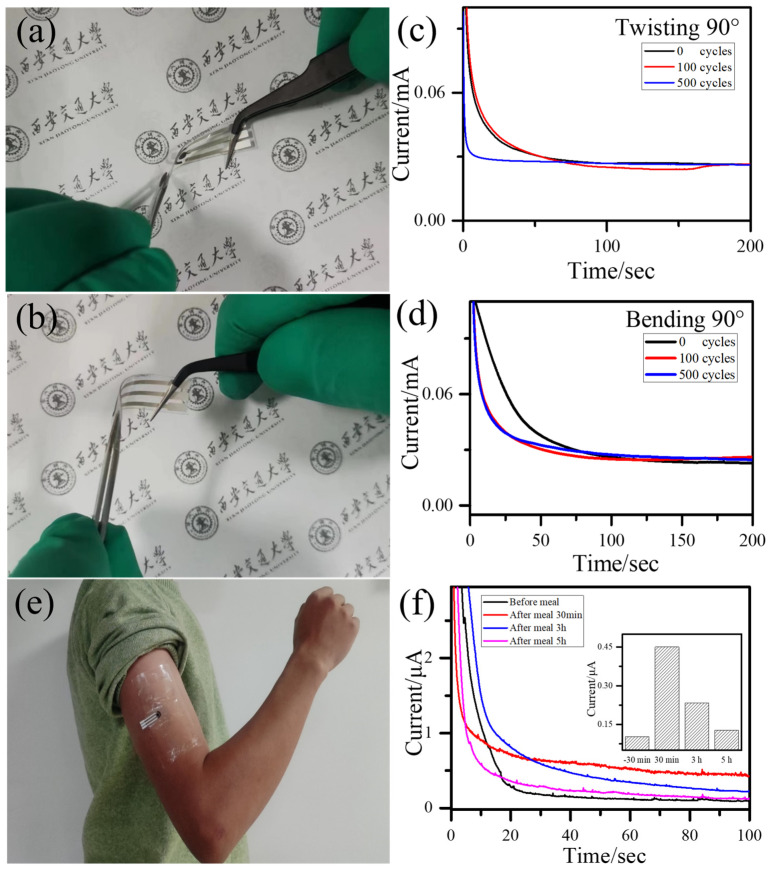
Optical images of (**a**) twisting 90°, (**b**) bending 90° and (**e**) human real sweat measuring method. Amperometric response of the composite films by 0.5 mM glucose concentration with (**c**) twisting and (**d**) bending 0, 100, and 500 cycle times. (**f**) Amperometric response of the sensor before meal (−30 min), after meal 30 min, 3 h and 5 h, the normalized amperometric response shown as insert graph.

**Table 1 sensors-23-00695-t001:** Analytical performances of Cu_x_O NFs/Cu NPs nanocomposites-based sensor compared with other non-enzymatic glucose sensors^α^.

Electrode	Sensitivity (μA mM^−1^ cm^−2^)	Liner Range (Up to mM)	Detection Limit (nM)	Reference
Cu/CuO/Carbon	320	3.3	7560	[15]
Cu/Cu_2_O/Carbon Spheres	63.8	0.69	5000	[23]
CuO Nanoflakes	431.3	2.5	80	[17]
CuNOx thin film	603.42	7	94,210	[37]
Cu-xCu_2_O/graphene film	230.86	10	16,000	[38]
PSS-CuO-CNSs/ graphite	663.2	10	230	[39]
CuO-screen-printed electrodes	383	3	10,000	[40]
Cu_x_O hollow spheres/rGO	635.315	14	1150	[41]
This work	779	>2.5	79.1	This work

## Data Availability

The data are not publicly available due to the relevant project regulations.

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
