# Peer review of "Wearable Noninvasive Glucose Sensor Based on CuxO NFs/Cu NPs Nanocomposites"

_sensors, 2023, doi:10.3390/s23020695_

Round 1
Reviewer 1 Report
Yu et.al synthesized the CuxO nanomaterials with a high surface area by the involvement of CuCl2 for wearable glucose sensing. The method is simple but can significantly increase the sensing performance. The materials are well characterized, and the sensing performance is inclusively investigated. The sensor can detect the low concentration of glucose in human sweat. However, there are still many points needed further clarification before acceptance.
1. Figure 1a The diagram is not clear, especially it is hard to see the shape of CuxO NFs/Cu NPs.
2. Some errors:
a)The Figure 2b has two white frames
b)Why do Figures 2c and 2d have the green frames but without the FFT of the lattice?
c)Figure 2f the baseline and fitting line with the same color
d)Figure S2d the material phases were not labled
3. Figure 3f should present error bars to show the repeatability of the sensor
4. The SEM images of the synthesized copper oxide nanoparticles with and without the CuCl2 have similar morphology. How does CuCl2 affect the growth of materials?
Author Response
Dear Editor,
We thank you very much for allowing us to revise our manuscript entitled “Wearable noninvasive glucose sensor based on CuxO NFs/Cu NPs nanocomposites” (sensors-2130952). We are grateful for your and the reviewer’ comments and have fully considered all the comments. We have revised the manuscript according to all the comments in the revised manuscript accordingly. Please see our point-to-point response to the reviews’ comments in the Response to Reviewers Letter. We hope, with these improvements based on your suggestions and the referees’ comments, the quality of our revised manuscript would meet the publication standard of Sensors.
The revisions have been done in the attached revised manuscript, and all the changes in the manuscript have been delineated in red. Please let me know if I can provide further information that would assist in your consideration of our manuscript. Many thanks.
Sincerely yours,
Chuncai Kong
MOE Key Laboratory for Non-Equilibrium Synthesis and Modulation of Condensed Matter, School of Physics, Xi’an Jiaotong University, No. 28, Xianning West Road, Xi’an, Shaanxi, 710049, P. R. China.
E-mail: kongcc@xjtu.edu.cn

Reviewer 2 Report
The results are interesting and I recommend its publication with minor revision.
1. Please discuss about prepared sensor surface area
2. Nafion gives very broad peak, how you justify about nanomaterial influence
3. What is drying time after immobilized with mixture
4. For drying purpose did used any methods or is it easily dried
5. Please include error bar for fig. 3f
6. What is the purity of these analytical samples?
7. Check the experimental section and write clearly the used conditions (volumes in mL, pH, concentration etc.) used in performing the experiments. This section should be clearly usable by readers.
8. Have the authors tried to fit the experimental data in using an equivalent circuit? How charge transfer resistance value is obtained?
9. Authors should provide the calculation for the surface coverage area.
10. In the real sample analysis what is the dilution factor?
11. Please mention sensitivity of the electrode
12. Chemicals purity missing
Author Response

(The authors gave the same response as above.)
